# Probabilistic Federated Prompt-Tuning with Non-IID and Imbalanced Data

**Pei-Yau Weng**
Washington State University
`pei-yau.weng@wsu.edu`

**Minh Hoang**
Princeton University
`minhhoang@princeton.edu`

**Lam M. Nguyen**
IBM Research
`lamnguyen.mltd@ibm.com`

**My T. Thai**
University of Florida
`mythai@cise.uf.edu`

**Tsui-Wei Weng**
University of California San Diego
`lweng@ucsd.edu`

**Trong Nghia Hoang**[*]
Washington State University
`trongnghia.hoang@wsu.edu`

## Abstract

Fine-tuning pre-trained models is a popular approach in machine learning for solving complex tasks with moderate data. However, fine-tuning the entire pre-trained model is ineffective in federated data scenarios where local data distributions are diversely skewed. To address this, we explore integrating federated learning with a more effective prompt-tuning method, optimizing for a small set of input prefixes to reprogram the pre-trained model's behavior. Our approach transforms federated learning into a distributed set modeling task, aggregating diverse sets of prompts to globally fine-tune the pre-trained model. We benchmark various baselines based on direct adaptations of existing federated model aggregation techniques and introduce a new probabilistic prompt aggregation method that substantially outperforms these baselines. Our reported results on a variety of computer vision datasets confirm that the proposed method is most effective to combat extreme data heterogeneity in federated learning.

## 1 Introduction

The proliferation of personal devices has transformed our data landscape into numerous federated systems with distinct resource constraints, data representations, distributions, and ownership. This has motivated the development of new machine learning (ML) paradigms that enable collaborative learning across different systems while respecting their data privacy. A prominent framework to substantiate this collaborative scheme is federated learning (FL), which allows multiple systems to train a common model without sharing their private data [1, 2, 3, 4, 5, 6, 7, 8, 9, 10, 11, 12].

Existing FL methods often assume that learning begins from scratch and does not build on prior expertise. On the other hand, fine-tuning pre-trained or foundation models [13] is an emerging paradigm for efficient generation of ML solutions. Almost all state-of-the-art models in natural language processing (NLP) are now fine-tuned versions of foundation models, such as BERT [14], BART [15], RoBERTa [16], and T5 [17]. Likewise, many top performing vision models have also benefited from the generalization capability of foundation models, such as the vision transformer model [18], which was trained on large-scale and generic datasets such as ImageNet [19]. Nonetheless, this fine-tuning practice has only recently been investigated in federated learning [20, 21].

Current research in this direction demonstrated various benefits of integrating fine-tuning into FL. For example, fine-tuning can utilize information better in decentralized data scenarios and thus improves the performance in various FL scenarios, mostly in NLP [22, 23, 24, 25]. Interestingly, it has been pointed out that even a simple initialization of local clients with a foundation model can prevent solution drift to some extent in large-scale scenarios with heterogeneous local data distributions [20, 21, 23]. This is also consistently observed in the results of our case studies in Fig. 1.

---

[*]Corresponding authors: Pei-Yau Weng, Minh Hoang, Trong Nghia Hoang.

38th Conference on Neural Information Processing Systems (NeurIPS 2024).

However, in FL environments that depend on frequent communication and synchronization of model updates across multiple devices, fine-tuning the entire pre-trained model is often infeasible due to limited local storage and communication bandwidth. Our study also reveals that full-model fine-tuning approaches fall short when the federated data are not only *heterogeneous* but also *imbalanced*. Our case studies in Fig. 1 in particular show that when the local data are both heterogeneous and imbalance, federated learning with full-model adaptation suffers a huge performance drop, highlighting its instability in extreme data settings.

To address the resource bottleneck, several works in the recent federated learning literature have turned to a new class of parameter-efficient tuning method called prompt tuning. This method focuses on engineering cues, such as extra tokens which are appended to the input embedding in a transformer architecture. Such tokens or prompts provide beneficial context to performing the computational task, similar to how hints can be provided to assist puzzle solving. Local sets of prompts can then be aggregated or personalized using existing federated learning strategies. For example, the prior works of [26] and [27] use FEDAVG [2] to aggregate the local prompts while [28] integrates prompt-tuning into personalized federated learning, which helps generate client-specific sets of prompts that are well-customized to the corresponding local data distribution.

Although federated prompt-tuning approaches eliminate the need to update and communicate hundreds of millions of network weights, there is a substantial gap between the performance of these approaches in highly heterogeneous, data-imbalanced settings, and the upper-bound performance of fine-tuning in the centralized data setting, as shown in Fig. 1. This gap can be attributed to the fact that locally learned prompt sets are learned in arbitrary orders, which are generally unaligned across clients. That is, the same prompt position across different clients might encode different contextual information about the local data. A simple aggregation that disregards prompt alignment might attempt to combine prompts from different contexts and collapse into less informative prompts. This has not been explicitly modeled in prior federated prompt-tuning work.

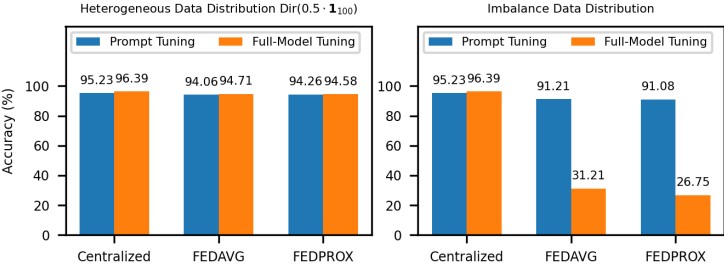

Figure 1: Test Accuracy (%) achieved on the CIFAR-10 dataset by solving Eq. (3) via centralizing data, using FEDAVG, and using FEDPROX on (orange) full-model (FM) and (blue) prompt-tuning (PT) setups. The evaluation is performed under (left) a standard (non-extreme) heterogeneous data partition; and (right) an extremely imbalanced data partitioning scheme (see Section 4).

To address this issue, we adopt a hierarchical probabilistic modeling approach to characterize both the generation and alignment of local prompts in this paper. We cast the server aggregation step as an alignment of local prompt sets. This alignment discovers and aggregates local prompts encoding similar contextual information into summarizing prompts. On the client side, we view each local prompt set as an independent sample drawn from a generative model parameterized by the global summarizing prompts. In this view, each local prompt can be seen as a probabilistic exploration initialized by a randomly selected summarizing prompt. The alignment between prompts can be characterized in terms of their association with the summarizing prompts, which can be inferred via learning the parameters of the above generative model. We summarize our main contributions below:

**C1.** We formulate the prompt summarizing procedure as a probabilistic set modeling task in which each local set is assumed to be an independent sample of a random point process. The alignment of similar prompts across different local sets can be set as part of the modeling parameterization, whose optimization can be interleaved with the optimization of the point process's parameters (Section 3.2).

**C2.** We develop an algorithm to find the most probable association between the local and summarizing prompts. This association is viewed as a latent variable in the above generative model. Specifically, we cast its inference as a classical weighted bipartite matching task via an interesting observation that the association between the summarizing prompt and each local set of prompts participate linearly to the loss function of our proposed generative model (Section 3.3).

**C3.** We compare the performance of our method against various federated prompt-tuning baselines based on existing heterogeneous federated learning techniques to demonstrate its effectiveness. Our reported results on a variety of experiments and baselines demonstrate consistently that our method is most effective to combat data imbalance in extreme heterogeneous scenarios (Section 4).

## 2 Related Work

Federated learning (FL) [1, 2] is a collaborative framework that allows multiple parties to collaborate on training a common model without sharing their private data. In FL, the training data are distributed across $m$ clients. The $t$-th client owns a local, private dataset $D_t = \{(\mathbf{x}_{tm}, y_{tm})\}_{m=1}^{n_t}$ comprising $n_t$ data points. The goal is to fit a model $\mathbf{w}_*$ using all private datasets without centralizing them. That is,

$$\mathbf{w}_* = \arg\min_{\mathbf{w}} \left\{ L(\mathbf{w}) \triangleq \sum_{t=1}^{m} \left(\frac{n_t}{n}\right) \cdot L_t(\mathbf{w}) \right\} , \text{ where } L_t(\mathbf{w}) \triangleq \frac{1}{n_t} \sum_{m=1}^{n_t} \ell(\mathbf{x}_{tm}, y_{tm}; \mathbf{w}) \quad (1)$$

and $n = n_1 + n_2 + \ldots n_m$ denotes the total number of data points while $\ell(\mathbf{x}_{tm}, y_{tm}; \mathbf{w})$ denotes some loss function of choice. Clients collaborate via sharing their local models instead of data. Each client can run multiple local updates before sharing its local model for aggregation. This helps reduce the number of communication rounds while still preserving the convergence guarantee. [2] names this the FEDAVG algorithm, which iterates between local optimization and global aggregation:

$$\mathbf{w}_t^{(r)} = U\left(\mathfrak{L}_t, \mathbf{w}_*^{(r-1)}\right) \forall t \in [m], \quad \mathbf{w}_*^{(r)} = \sum_{t=1}^{m} \left(\frac{n_t}{n}\right) \cdot \mathbf{w}_t^{(r)} . \quad (2)$$

The local update routine $U(\mathfrak{L}_t, .)$ is typically standard gradient updates such as SGD or Adam. At the beginning of an iteration, the local weight is set to be the global estimate from the previous communication round. In practice, local data distributions tend to diverge which consequently causes the local updates in Eq. (2) to have different convergence points across clients. This is known as the solution drift phenomenon, which often decreases the performance of the federated model. Numerous approaches had been proposed to mitigate this phenomenon, which includes the following directions:

**Client Regularization.** These methods focus on augmenting the local update strategies to prevent clients from drifting apart. For example, FEDPROX [29] introduces an $\ell_2$-regularization term to penalize updates that diverge the local model from the global model. FEDDYN [30] uses a FL-adapted version of [31] on distributed optimization as the regularizer. SCAFFOLD [32] attempts to correct the direction of local gradient updates at each client using their sum of gradients. [33] utilizes the similarity between model representations to correct local training via contrastive learning.

**Server Regularization.** Another approach to prevent the drifting effect caused by heterogeneous local distributions is to replace the model average in Eq. (2) with a different mechanism of model aggregation. For example, [7, 6] decomposes client models into bags of neurons and performs a non-parametric clustering of neurons. Cluster centroids are used to synthesize an aggregated model for the next communication round. This approach bypasses the drifting phenomenon as the number of cluster is non-parametric and can be adjusted to accommodate new information patterns. In a similar vein, [34] adopts a probabilistic perspective of model aggregation, sampling higher-quality global models and combining them via Bayesian model ensemble, leading to a more robust aggregation. [35] presents a data-free knowledge distillation methods for FL that help to train generators without compromising clients' data. [36] redistributes each client's shared model to others for $b$ consecutive communication iterations, exposing it to various heterogeneous data sources before performing aggregation, curving their solution divergence.

**Personalization.** Instead of learning a single, universal model, [29, 37, 38, 39, 40] seek to learn personalized models for all clients. [3, 39] formulates this problem as multi-task learning, whereas [37] and [38] respectively ground it in meta-learning and regularization frameworks. Several recent works impose a shared data representation across clients [40, 41, 42, 43, 44], which is optimized using existing FL techniques, and locally fine-tune a small classification head to achieve personalization. However, personalized models trained in this manner tend to only perform well on its corresponding local test set, and not on a comprehensive test set combining all local test data (see Section 4).

Finally, most previous work assume that each FL client has sufficient data to adequately train its local model. This often contradicts real-world situations where data are scarce [45, 36, 46, 47, 48]. While data scarcity is not a new challenge of machine learning, it has not been thoroughly investigated in the

context of FL. [36] points out that with limited, scarce data, local models often have bad qualities, and aggregating such models tend to result in poor global performance. Although fine-tuning pre-trained large models is an increasingly popular technique to combat data shortage, most existing FL works (including [36] which aims to address data scarcity) have not tried to leverage this resource. This motivates us to investigate prompt-tuning as a new FL paradigm.

# 3 Probabilistic Federated Prompt-Tuning

First, Section 3.1 provides a concise background on the standard prompt tuning technique in FL setting. Motivated by the result of our case study (see Fig. 1), Section 3.2 further introduces a probabilistic federated prompt-tuning framework that aims to close up the performance gap between federated prompt-tuning and centralized full-model fine-tuning in data imbalance settings. Our framework characterizes each local model as a random set of prompts distributed by a hierarchical generative model. An effective optimization algorithm to learn these is then detailed in Section 3.3. An overall diagram featuring a bird-view of our framework is also provided in Fig. 2.

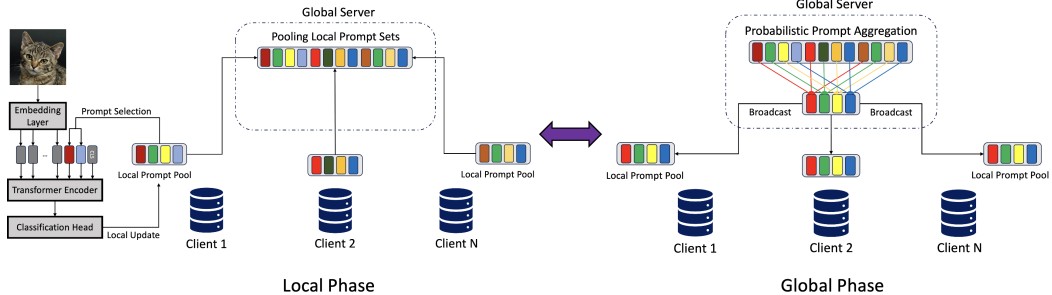

Figure 2: Workflow of Probabilistic Federated Prompt Aggregation: (left) each client selects a subset of prompts from the global set of summarizing prompts using the prompt-selection mechanism adapted from [49], and fine-tune them using local data; and (right) the server collects all local prompt sets and updates the global summarizing prompts that aggregate similar local prompts. This is achieved by our proposed probabilistic federated prompt aggregation (PFPT) algorithm.

## 3.1 Prompt-Tuning with Pre-Trained Model

We use a pre-trained Vision Transformer [18] in all our experiments. The pre-trained model is a composition $F_c \circ F_a \circ F_e$ where $F_c$ is the classification head, $F_a$ is the stack of attention blocks, and $F_e$ is the input embedding block. Keeping both $F_a$ and $F_e$ frozen, the local solution for each client $t$ can be represented as $F_{c,t} \circ F_a \circ (F_e \cup \boldsymbol{\omega})$ which applies a personalized prediction head on the output of the frozen attention block $F_a$ whose (set) inputs are the union of $F_e$ and a set $\boldsymbol{\omega}$ of reprogramming prompts [49]. $\boldsymbol{\omega}$ is in turn a union of $k$ individual prompts $\boldsymbol{\omega} = \boldsymbol{\omega}_1 \cup \boldsymbol{\omega}_2 \cup \ldots \cup \boldsymbol{\omega}_k$. Each local solution is characterized by a tuple of $(F_{c,t}, \boldsymbol{\omega})$ comprising a personalized head $F_{c,t}$ and common prompt set $\boldsymbol{\omega}$. The FL framework in Eq. (1) is now re-cast as

$$\boldsymbol{\omega}_* = \arg\min_{\boldsymbol{\omega}} \left\{ \sum_{t=1}^{m} \left( \frac{n_t}{n} \right) \cdot L_t(\boldsymbol{\omega}) \right\} \text{ with } L_t(\boldsymbol{\omega}) \triangleq \max_{F_{c,t}} \left\{ \left( \frac{1}{n_t} \right) \cdot \sum_{m=1}^{n_t} \ell\left(\mathbf{x}_{tm}, y_{tm}; F_{c,t}, \boldsymbol{\omega}\right) \right\} \text{ (3)}$$

where $\ell(\mathbf{x}_{tm}, y_{tm}; F_{c,t}, \boldsymbol{\omega}) = \ell(\mathbf{x}_{tm}, y_{tm}; F_{c,t} \circ F_a \circ (F_e \cup \boldsymbol{\omega}))$ which makes explicit the leverage of the existing (pre-trained) expertises $(F_a, F_e)$. The optimal solution $\boldsymbol{\omega}_*$ of Eq. (3) can be approximated using any of the existing federated learning algorithms, such as FEDAVG [2] and FEDPROX [29].

**Remark.** There is an alternative approach to enabling light-weight federated fine-tuning. Instead of using prompts, approaches in this direction use a (learnable) adapter network that adapts the output of an intermediate (frozen) neural network segment before passing it to the rest of the (frozen) neural network. Local adapters across client can then be aggregated using FEDAVG [21]. This direction is however orthogonal to federated prompt-tuning, which is our main focus here. Further investigation into probabilistic methods for adapter aggregation would be a potential follow-up of our current work.

## 3.2 Probabilistic Prompt Model

Intuitively, our federated prompt tuning framework is an analog to traditional FL on the space of prompts. Prior to each communication round, we suppose that each local client $t$, via prompt-tuning,

has obtained a set of $n_t$ prompts $\boldsymbol{\omega}_t \triangleq \{\boldsymbol{\omega}_{t1}, \boldsymbol{\omega}_{t2}, \ldots, \boldsymbol{\omega}_{tn_t}\}$. As the participating clients upload their local prompt sets, the server aggregates the combined set and subsequently returns a set of $n$ summary prompts, denoted as $\boldsymbol{\Phi} \triangleq \{\boldsymbol{\phi}_i\}_{i=1}^n$. We will elaborate on this aggregation step in Section 3.2.2. Similar to standard FL, this aggregated set is also distributed to all clients at the beginning of the next communication round. Each client then selects the most relevant prompts from this set for further fine-tuning. This sampling step follows a generative process described in Section 3.2.1.

### 3.2.1 Generative Model

At the beginning of each communication round, each client constructs a set of $n_t$ prompt initializations given the server-broadcast summary $\boldsymbol{\phi}$. We model this construction by a random generative process. First, each local prompt initialization, $\boldsymbol{\omega}_{tk}$, is modeled as a sample drawn from a Gaussian with learnable neural parameterization,

$$\boldsymbol{\omega}_{tk} \quad \sim \quad \mathbb{N}\Big(\boldsymbol{\psi}_{tk}, \mathrm{diag}\Big(\alpha(\boldsymbol{\psi}_{tk}; \boldsymbol{\gamma})\Big)\Big), \tag{4}$$

where the diagonal covariance matrix are parameterized as the output of a neural net $\alpha$ (with weight $\gamma$) on the mean parameter $\boldsymbol{\psi}_{tk}$. The set of mean parameters $\boldsymbol{\psi}_t \triangleq \{\boldsymbol{\psi}_{tk}\}_{k=1}^{n_t}$ is in turn modeled as a random subset of the (server-broadcast) summarizing prompts $\boldsymbol{\phi}$, following a Bernoulli point process prior with a finite mixture as the base measure:

$$Q_t(\boldsymbol{\psi}) \quad \sim \quad \mathrm{BeP}\left(\sum_{i=1}^n \sigma\Big(g(\boldsymbol{\phi}_i; \mathbf{w})\Big) \cdot \mathbb{I}\Big(\boldsymbol{\psi} = \boldsymbol{\phi}_i\Big)\right). \tag{5}$$

Here, $\sigma$ denotes the sigmoid activation function, and $g$ is another deep-parameterized function parameterized by weight $\mathbf{w}$. By definition of the Bernoulli process, the sampled measure is given by:

$$Q_t(\boldsymbol{\psi}) = \sum_{i=1}^n c_i \cdot \mathbb{I}\Big(\boldsymbol{\psi} = \boldsymbol{\phi}_i\Big), \tag{6}$$

where $c_i \in \{0, 1\}$ is the outcome of a Bernoulli trial with bias $\sigma(g(\boldsymbol{\phi}_i; \mathbf{w}))$, that is:

$$\mathbb{P}_i(c_i) \quad = \quad \sigma\Big(g(\boldsymbol{\phi}_i; \mathbf{w})\Big)^{c_i} \cdot \Big(1 - \sigma\Big(g(\boldsymbol{\phi}_i; \mathbf{w})\Big)\Big)^{1-c_i}. \tag{7}$$

In layman term, this process simply means we will observe the result of a coin toss $c_i$ for every summarizing prompt $\boldsymbol{\phi}_i$. If this coin lands on head ($c_i = 1$), the summarizing prompt $\boldsymbol{\phi}_i$ will be included in the set of mean parameters. Vice versa, if this coin lands on tail, we will skip $\boldsymbol{\phi}_i$. Because of this, we remark that $n_t$ could be different from the number of prompts in the previous round.

### 3.2.2 Prompt Aggregation

Given the above generative story, we can now describe our algorithm to find the set of summarizing prompts $\{\boldsymbol{\phi}_i\}_{i=1}^n$, which maximize the likelihood of observing local prompts $\{\boldsymbol{\omega}_t\}_{t=1}^m$. The resulting optimal set of summarizing prompts can then be used to re-program the pre-trained model, which correspond to the global fine-tuned solution. Assuming that each prompt captures a particular fine-tuning pattern or concept, summarizing the prompt sets characterizing the local solutions will allow local models to be aggregated on the concept level, naturally mitigating the solution drift effect caused by compounding impact of data heterogeneity and pre-trained weight interference. This is substantiated in the overall computation process below.

Let $z_{tk}^i \in \{0, 1\}$ denotes whether the local prompt $\boldsymbol{\omega}_{tk}$ was sampled from a Gaussian centered at the $i$-th summarizing prompt $\boldsymbol{\phi}_i$. That is, the assignment variable $z_{tk}^i = 1$ if and only if there exists $k \in [n_t]$ such that $\boldsymbol{\psi}_{tk} = \boldsymbol{\phi}_i$. Each client $t$ can now be represented as $(\boldsymbol{\omega}_t, \mathbf{z}_t)$ where $\mathbf{z}_t = \{\mathbf{z}_t^i\}_{i=1}^n$, $\mathbf{z}_t^i = (z_{t1}^i, z_{t2}^i, \ldots, z_{tn_t}^i)$ and $\boldsymbol{\omega}_t = (\boldsymbol{\omega}_{t1}, \boldsymbol{\omega}_{t2}, \ldots, \boldsymbol{\omega}_{tn_t})$.

Now, using the generative process specified in the previous section and all its defining parameters $\boldsymbol{\varphi} = (\mathbf{w}, \boldsymbol{\gamma}, \{\boldsymbol{\phi}_i\}_{i=1}^n)$, the log likelihood of each client can be derived as follow,

$$\log \mathbb{P}\left(\boldsymbol{\omega}_t, \mathbf{z}_t \mid \boldsymbol{\varphi}\right) \quad = \quad \log \mathbb{P}\left(\boldsymbol{\omega}_t \mid \mathbf{z}_t, \boldsymbol{\varphi}\right) + \log \mathbb{P}\left(\mathbf{z}_t \mid \boldsymbol{\varphi}\right), \tag{8}$$

where each of the summand on the right-hand side is computed below. First, given $(\mathbf{z}_t, \boldsymbol{\varphi})$,

$$\mathbb{P}\Big(\boldsymbol{\omega}_t \mid \mathbf{z}_t, \boldsymbol{\varphi}\Big) = \prod_{k=1}^{n_t} \mathbb{N}\Big(\boldsymbol{\omega}_{tk} \mid \boldsymbol{\psi}_{tk}, \mathrm{diag}\Big(\alpha\Big(\boldsymbol{\psi}_{tk}; \boldsymbol{\gamma}\Big)\Big)\Big) \text{ with } \boldsymbol{\psi}_{tk} = \Big(z_{tk}^1 \cdot \boldsymbol{\phi}_1 + \ldots + z_{tk}^n \cdot \boldsymbol{\phi}_n\Big) \tag{9}$$

---

**Algorithm 1** Probabilistic Federated Prompt Tuning (PFPT)

---

**input**: pre-trained model $F$, no. $\tau$ of iterations, no. $m$ of sampled clients per iteration
**output**: optimized set of prompts $\mathbf{\Phi}$

 1: initialize global summarizing prompts $\mathbf{\Phi}$
 2: **for** $s = 1$ **to** $\tau$ **do**
 3:     sample $m$ clients with private datasets $\{D_t\}_{t=1}^m$
 4:     **for** $t = 1$ **to** $m$ **do**
 5:         $\boldsymbol{\omega}_t \leftarrow \text{client}(F, D_t, \mathbf{\Phi})$ // local prompt-tuning – optimizing $L_t(\boldsymbol{\omega}_t)$ in Eq. (3)
 6:     **end for**
 7:     $\mathbf{\Phi}, \mathbf{z} \leftarrow \text{server}(\{\boldsymbol{\omega}_t\}_{t=1}^m)$ // solving Eq. (11)
 8:     $\mathbf{\Phi} \leftarrow \left\{ \phi_i \in \mathbf{\Phi} \mid \sum_{t=1}^m \sum_{k=1}^{n_t} z_{tk}^i > 0 \right\}$ // remove inactive prompts
 9: **end for**
10: **return** the set $\mathbf{\Phi}$ of optimal prompts

---

Then, given $\boldsymbol{\varphi}$ and let $c_{ti} = z_{t1}^i + z_{t2}^i + \ldots + z_{tn_t}^i$,

$$\mathbb{P}\Big(\mathbf{z}_t \mid \boldsymbol{\varphi}\Big) = \prod_{i=1}^n \left( \sigma\Big(g(\phi_i; \mathbf{w})\Big)^{c_{ti}} \right) \times \prod_{i=1}^n \left( \Big(1 - \sigma\Big(g(\phi_i; \mathbf{w})\Big)\Big)^{\big(1 - c_{ti}\big)} \right) \tag{10}$$

Eq. (9) follows from the Gaussian likelihood of local prompt in Eq. (4) with the additional identification form for $\psi_{tk}$ which is computable if $\mathbf{z}_t$ is given. Eq. (10) on the other hand is the consequence of the Bernoulli process in Eq. (6), which might not be immediately straight-forward. Its detailed derivation will be given in Appendix B.

### 3.2.3 Overall Workflow

Using Eq. (9) and (10), we have the following prompt aggregation formulation,

$$\mathbf{\Phi}^* \quad = \quad \underset{\mathbf{\Phi}}{\arg\max} \left\{ \max_{\mathbf{w}, \boldsymbol{\gamma}, \mathbf{z}} \sum_{t=1}^m \log \mathbb{P}\Big(\boldsymbol{\omega}_t, \mathbf{z}_t \mid \mathbf{w}, \boldsymbol{\gamma}, \mathbf{\Phi}\Big) \right\} \tag{11}$$

where $\mathbf{\Phi} = \{\phi_i\}_{i=1}^n$, $\mathbf{z} = \{\mathbf{z}_t\}_{t=1}^m$, and $\boldsymbol{\gamma}$ is previously defined in Eq. (4). The probability term on the right-hand side can be further expanded into computable terms using Eq. (8), Eq. (9), and Eq. (10). Solving Eq. (11) is therefore the key step in our federated fine-tuning framework, whose overall flow is summarized in Alg. 1. Our algorithm proceeds in multiple iterations. At each iteration, a set of $m$ clients is sampled. Each client selects the most relevant subset of summarizing prompts from the server using a mechanism adapted from continual visual prompt-tuning [49]. The selected subset of summarizing prompts are then fine-tuned using local data (see lines 3-4). The sets of fine-tuned prompts are subsequently uploaded to the server, which performs the aggregation via solving Eq. (11) (see line 6). The aggregation step has two subtleties: (1) the total number of global prompts, $n = n_1 + \ldots + n_t$, is proportional to the total number of prompts across clients, making the entire algorithm non-parametric; and (2) once the optimal assignment parameters $\mathbf{z}$ are found via solving Eq. (11), global prompts that were not assigned to any client will be removed (see lines 7 and 8). Intuitively, if the sampled clients are similar, their prompt sets will substantially overlap, which reduces the total number of distinct prompts selected by all clients. Otherwise, if the sampled clients are dissimilar, there will be less overlapping and the union set of selected global prompts will expand, increasing the complexity of the resulting model to match the increased data heterogeneity.

### 3.3 Optimization

Solving Eq. (11) above is however not trivial due to its mixed set of discrete/continuous variables. To sidestep this intractability, we instead formulate it as a bi-level optimization that alternates between two sub-tasks: (1) optimizing $\mathbf{z}$ given $\mathbf{w}, \boldsymbol{\gamma}$ and $\mathbf{\Phi}$; and (2) optimizing $\mathbf{\Phi}, \mathbf{w}, \boldsymbol{\gamma}$ given $\mathbf{z}$. Among which, solving sub-task (2) is straight-forward since it only requires being able to differentiate the expression in Eq. (9) and Eq. (10), which is trivial if we use the current estimation of the (discrete) assignment parameters $\mathbf{z}$. On the other hand, solving sub-task (1) is seemingly prohibitive expensive because it involves optimizing the (discrete) assignment parameters $\mathbf{z}$. Fortunately, this can be

mitigated by casting both Eq. (9) and Eq. (10) into a linear form with respect to the assignment parameter $\mathbf{z}$. Intuitively, it might be confusing why such linear form can be achieved at all given that $\mathbf{z}$ is input to a non-linear log probability function in Eq. (9) and Eq. (10). However, we must recognize that for each client $t$, there is at most one local prompt from $t$ which can be associated with the $i^{\text{th}}$ summarizing prompt. This implies $\mathbf{z}_t^i \triangleq (z_{t1}^i, z_{t2}^i, \ldots, z_{tn_t}^i)$ is either a zero or one-hot vector. This is an important observation because a linear function of such zero or one-hot vector will remain linear regardless of any (non-linear) post-processing transformation. It is shown in Lemma 3.1 below.

**Lemma 3.1.** *For any scalar function $g(\mathbf{r})$ and a binary vector $\boldsymbol{\zeta} = [\zeta_1, \zeta_2, \ldots, \zeta_n]$ such that $\zeta_i \in \{0,1\}$ and $\boldsymbol{\zeta}$ has at most one non-zero component, we have*

$$g\left(\sum_{i=1}^n \zeta_i \cdot \mathbf{r}_i\right) = \sum_{i=1}^n \left(\zeta_i \cdot g(\mathbf{r}_i)\right) \tag{12}$$

*with respect to any set $\{\mathbf{r}_i\}_{i=1}^n$ of valid inputs to $g(\mathbf{r})$.*

The proof of Lemma 3.1 is straight-forward. First, if there is no non-zero component, both sides of Eq. (12) evaluate to $g(0)$. Otherwise, suppose the only non-zero component appears at position $\kappa$, both sides of Eq. (12) will evaluate to $g(\mathbf{r}_\kappa)$. In both cases, Eq. (12) holds. Using Lemma 3.1, we can establish the desired linear forms for Eq. (10) and Eq. (11) which are its immediate consequences.

**Lemma 3.2.** *Let $\mathbb{P}(\boldsymbol{\omega}_t \mid \mathbf{z}_t, \boldsymbol{\varphi})$ defined as in Eq. (9). Let $L_1(\mathbf{z}) = \sum_{t=1}^m \log \mathbb{P}(\boldsymbol{\omega}_t \mid \mathbf{z}_t, \boldsymbol{\varphi})$, considering $(\boldsymbol{\omega}_t, \boldsymbol{\varphi})$ as constants. We have*

$$L_1(\mathbf{z}) = \sum_{i=1}^n \sum_{t=1}^m \sum_{k=1}^{n_t} z_{tk}^i \cdot \log \mathbb{N}\left(\boldsymbol{\omega}_{tk} | \boldsymbol{\phi}_i, \operatorname{diag}\left(\alpha\left(\boldsymbol{\phi}_i; \boldsymbol{\gamma}\right)\right)\right) \tag{13}$$

*which is linear in terms of the assignment parameter $\mathbf{z}$.*

**Lemma 3.3.** *Let $\mathbb{P}(\mathbf{z}_t \mid \boldsymbol{\varphi})$ defined as in Eq. (10). Let $L_2(\mathbf{z}) = \sum_{t=1}^m \log \mathbb{P}(\mathbf{z}_t \mid \boldsymbol{\varphi})$, considering $\boldsymbol{\varphi}$ as constants. Then, we have*

$$L_2(\mathbf{z}) = \sum_{i=1}^n \sum_{t=1}^m \sum_{k=1}^{n_t} z_{tk}^i \cdot \log\left(\frac{\sigma\left(g(\boldsymbol{\phi}_i; \mathbf{w})\right)}{1 - \sigma\left(g(\boldsymbol{\phi}_i; \mathbf{w})\right)}\right) + \sum_{i=1}^n \sum_{t=1}^m \log\left(1 - \sigma\left(g(\boldsymbol{\phi}_i; \mathbf{w})\right)\right) \tag{14}$$

*which is linear in terms of the assignment parameter $\mathbf{z}$.*

The detailed proof for Lemma 3.2 and Lemma 3.3 are deferred to Appendix C. Using these results, we can put together the overall optimization task for $\mathbf{z}$ while fixing the rest of the parameterization

$$\mathbf{z}^* = \underset{\mathbf{z}}{\operatorname{argmax}}\left\{\sum_{t=1}^m \log \mathbb{P}\left(\boldsymbol{\omega}_t, \mathbf{z}_t \mid \boldsymbol{\varphi}\right)\right\} = \underset{\mathbf{z}}{\operatorname{argmax}}\left\{L_1(\mathbf{z}) + L_2(\mathbf{z})\right\} \tag{15}$$

which is a weighted linear optimization task. The second equality follows from Eq. 8 and the results of Lemma 3.2 and Lemma 3.3. Now, if we further choose to optimize $\mathbf{z}_t$ iteratively while fixing in addition $\mathbf{z}_{-t}$, Eq. (15) reduces to a weighted bipartite matching task, which can be solved effectively in $\mathbb{O}((\max_{t=1}^m n_t)^3)$ processing time using the Hungarian algorithm [50]. A detailed pseudo-code implementing the above scheme is deferred to Appendix D.

## 4 Empirical Results

This section presents our empirical studies on a variety of computer vision datasets, including CIFAR-10 and CIFAR-100 [51], TinyImageNet [52] and a synthetic, diverse dataset created by pooling together the MNIST-M [53], Fashion-MNIST [54], CINIC-10 [55] and MMAFEDB (available on Kaggle) datasets, which is referred to as the 4-dataset.

Our experiments are conducted on two data settings: (a) Dirichlet-based heterogeneous partition following a previous setup in [6]; and (b) manual imbalanced data partition, as detailed below.

**Heterogeneous Partition.** We partition the training data into $m$ subsets (for $m$ clients). Each client has observations of all classes but the distributions of classes across clients are different. We simulate this using a $\operatorname{Dirichlet}(\alpha \cdot \mathbf{1}_s)$ distribution over an $s$-dimensional simplex where $s$ is the number of classes and $\alpha$ is the concentration parameter.

Table 1: Accuracy (%) achieved on the CIFAR-10 dataset by PFPT and other baselines.

|  | FEDAVG-PT | FEDPROX-PT | SCAFFOLD-PT | FEDOPT-PT | PFEDPG | GMM-PT | PFPT (ours) |
|---|---|---|---|---|---|---|---|
| $\alpha = 0.5$ | $94.06 \pm 0.45$ | $94.26 \pm 0.34$ | $92.26 \pm 0.18$ | $90.70 \pm 0.54$ | $88.00 \pm 0.22$ | $92.63 \pm 0.16$ | $\mathbf{94.39 \pm 0.51}$ |
| $\alpha = 0.1$ | $93.05 \pm 0.03$ | $93.05 \pm 0.23$ | $91.82 \pm 0.34$ | $88.20 \pm 0.39$ | $77.25 \pm 0.23$ | $92.68 \pm 0.28$ | $\mathbf{93.39 \pm 0.22}$ |
| imbalance | $91.21 \pm 0.16$ | $91.08 \pm 0.25$ | $87.06 \pm 1.27$ | $79.32 \pm 1.73$ | $43.16 \pm 0.27$ | $90.50 \pm 0.17$ | $\mathbf{91.45 \pm 0.08}$ |

Table 2: Accuracy (%) achieved on the CIFAR-100 dataset by PFPT and other baselines.

|  | FEDAVG-PT | FEDPROX-PT | SCAFFOLD-PT | FEDOPT-PT | PFEDPG-PT | GMM-PT | PFPT (ours) |
|---|---|---|---|---|---|---|---|
| $\alpha = 0.5$ | $79.40 \pm 0.27$ | $79.21 \pm 0.31$ | $70.83 \pm 0.18$ | $59.58 \pm 1.13$ | $43.47 \pm 0.39$ | $77.50 \pm 0.12$ | $\mathbf{80.24 \pm 0.24}$ |
| $\alpha = 0.1$ | $74.50 \pm 0.46$ | $73.60 \pm 0.38$ | $69.94 \pm 0.63$ | $60.31 \pm 0.39$ | $29.46 \pm 0.80$ | $70.49 \pm 0.46$ | $\mathbf{75.08 \pm 0.51}$ |
| imbalance | $70.07 \pm 0.35$ | $70.67 \pm 0.25$ | $69.35 \pm 0.96$ | $58.86 \pm 0.77$ | $11.57 \pm 0.25$ | $66.95 \pm 0.46$ | $\mathbf{72.05 \pm 0.93}$ |

Table 3: Accuracy (%) achieved on the TinyImageNet dataset by PFPT and other baselines.

|  | FEDAVG-PT | FEDPROX-PT | SCAFFOLD-PT | FEDOPT-PT | PFEDPG | GMM-PT | PFPT (ours) |
|---|---|---|---|---|---|---|---|
| $\alpha = 0.5$ | $86.38 \pm 0.18$ | $86.05 \pm 0.47$ | $78.70 \pm 0.33$ | $61.55 \pm 0.28$ | $50.92 \pm 0.44$ | $84.73 \pm 0.20$ | $\mathbf{86.91 \pm 0.14}$ |
| $\alpha = 0.1$ | $78.58 \pm 0.57$ | $79.19 \pm 0.28$ | $78.02 \pm 0.33$ | $63.81 \pm 0.79$ | $34.34 \pm 0.10$ | $76.90 \pm 0.28$ | $\mathbf{82.31 \pm 0.26}$ |
| imbalance | $75.21 \pm 0.73$ | $75.49 \pm 0.38$ | $76.88 \pm 0.45$ | $64.33 \pm 0.61$ | $11.92 \pm 0.13$ | $73.85 \pm 0.36$ | $\mathbf{78.21 \pm 1.25}$ |

Our experiments are conducted with $\alpha = 0.1$ and $\alpha = 0.5$. For each client $r$, we drawn a sample $\mathbf{p}_r \sim \text{Dirichlet}(\alpha \cdot \mathbf{1}_s)$ where $\mathbf{p}_{r,c} \in (0, 1)$ specifies the percentage of examples in class $c$ to be assigned to client $r$. For the CIFAR-10, CIFAR-100, and TinyImageNet datasets, we set $m = 100$. For the 4-dataset, we simulate 20 partitions for each of the 4 sub-dataset using $\text{Dirichlet}(\alpha \cdot \mathbf{1}_s)$ with $s = 10, 10, 10,$ and 7 for MNIST-M, FASHION-MNIST, CINIC-10, and MMAFEDB. This amounts to a total of 80 clients with diverse and heterogeneous data distribution.

**Imbalance Partition.** For the CIFAR-10 (10-class), CIFAR-100 (100-class) and TinyImageNet (200-class) datasets, the training data of each client is set to be dominated by a particular subset of classes, which amounts to $10\%$ of the total number of classes. $99\%$ of the local dataset of each client is set to belong to a certain $10\%$ of the total number of classes. For our synthetic 4-dataset, we partition each of the 4 sub-dataset into 20 subsets. Each subset is set so that $99\%$ of its data points belong to a single class. The remaining $1\%$ data of each class are then pooled together and evenly distributed among all clients. This amounts to 80 clients with extremely imbalance local datasets.

The above schemes are applied only on the train partition of each dataset. We use the default train/test partition for CIFAR-10, CIFAR-100, and TinyImageNet. For the 4-dataset, we sample 30K data points from the default train partition of each of its 4 sub-datasets. We also sample 2.5K data points from the default test partition of each sub-dataset. This amounts to a synthetic dataset with 120K data points in the train partition and 10K data points in the test partition.

For each data setup on each dataset, we compare the performance of our probabilistic federated prompt-tuning (PFPT) algorithm with those of a representative set of state-of-the-art federated learning algorithms adapted to the prompt-tuning setting in Eq. (3), which include FEDAVG [2], FEDPROX [29], SCAFFOLD [32], FEDOPT [56] and PFEDPG [28]. PFEDPG is a personalization method that is originally measured based on how well its personalized models perform on their corresponding local test sets. In this context, the performance target is however set on a global test set so we use PFEDPG's common model for evaluation. We will refer to these adapted baselines as FEDAVG-PT, FEDPROX-PT, SCAFFOLD-PT, FEDOPT-PT, and PFEDPG. In addition, we also compare PFPT against a simple prompt clustering baseline, which replaces the prompt averaging of FEDAVG by a Gaussian Mixture Model (GMM) clustering in which the cluster centroids are returned as the aggregated prompts. We refer to this baseline as GMM-PT. All results are averaged over 5 independent runs and reported in Tables 1 to 4 below. The result of each run is evaluated on a global test set, which comprises the entire test partition.

## 4.1 Non-IID Data with Locally Skewed Class Distributions

Table 1 reports the performance of PFPT and various federated prompt-tuning baselines on the CIFAR-10 dataset. In all three settings (i.e., heterogeneous partitions with $\alpha = 0.5$ and 0.1, as well as the imbalance partition), PFPT consistently achieves the best performance. On average, the classification accuracy of our method improves by $0.3\%$ over the closest competitors, FEDAVG-PT and FEDPROX-PT. While this gain seems modest, we note that the prompt tuning performance tends to fall off more significantly on several FL frameworks that were devised to counteract the effect of imbalanced data, such as SCAFFOLD-PT and FEDOPT-PT. In fact, the accuracy our method

Table 4: Accuracy (%) achieved on the synthetic 4-dataset, which combines data from MNIST-M [53], Fashion-MNIST [54], CINIC-10 [55], and MMAFEDB[1], by PFPT and other baselines.

| | FEDAVG-PT | FEDPROX-PT | SCAFFOLD-PT | FEDOPT-PT | PFEDPG | GMM-PT | PFPT (ours) |
|---|---|---|---|---|---|---|---|
| $\alpha = 0.5$ | $59.48 \pm 0.59$ | $59.50 \pm 0.70$ | $54.51 \pm 1.56$ | $40.69 \pm 2.29$ | $30.54 \pm 0.50$ | $57.05 \pm 1.04$ | $\mathbf{76.89 \pm 0.17}$ |
| $\alpha = 0.1$ | $48.74 \pm 0.63$ | $48.71 \pm 2.13$ | $46.65 \pm 3.17$ | $43.51 \pm 0.78$ | $25.40 \pm 0.17$ | $46.82 \pm 2.80$ | $\mathbf{70.29 \pm 0.32}$ |
| imbalance | $39.28 \pm 0.55$ | $39.65 \pm 1.89$ | $23.79 \pm 1.24$ | $44.43 \pm 1.78$ | $22.54 \pm 0.32$ | $41.37 \pm 1.12$ | $\mathbf{62.23 \pm 1.02}$ |

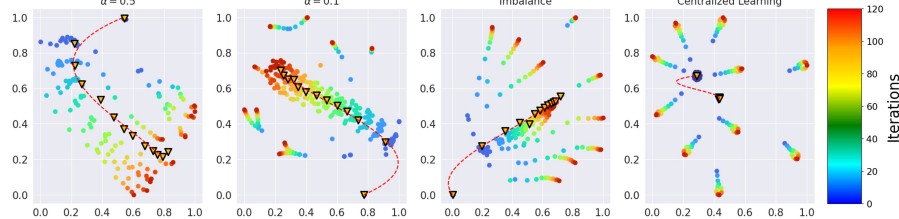

Figure 3: t-SNE plots of the (learned) summarizing prompts of PFPT on CIFAR-100 over 120 communication iterations with different heterogeneity settings. Yellow triangles denote the centroids of the t-SNE embeddings of the prompts. The dashed red line visualizes their trajectories.

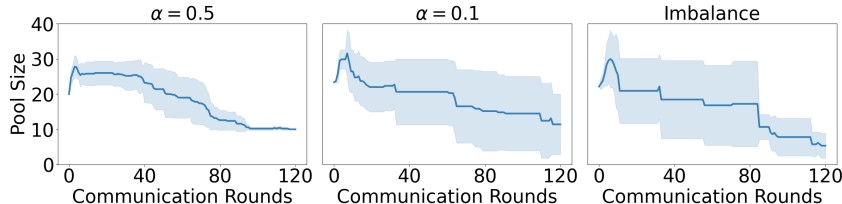

Figure 4: Variations in CIFAR-100 global prompt pool size across 120 communication rounds under different heterogeneity settings.

respectively improves by 2.7% and 7.0% over these baselines. This result clearly suggests that prompt tuning performance is more vulnerable to the heterogeneity challenge in FL, and thus requires a more specialized treatment. Finally, we observe that PFEDPG achieves the worst performance among all baselines (23.6% worse than PFPT on average). This performance gap is especially pronounced on the imbalance setting. While PFEDPG claimed to deal with both heterogeneity and prompt-tuning, this result is not surprising since it assumes that the local test sets are partitioned similarly to the local training sets, whereas our results are obtained on a holistic global test set.

We further repeat this experiment on the CIFAR-100 and TinyImageNet datasets, which are both more challenging than CIFAR-10 due to the increased number of classes (i.e., 100 and 200 classes respectively). The results of these experiments are reported in Table 2 and 3. As expected, our method still consistently outperforms other baselines. However, the performance gap between PFPT and other methods tends to widen proportionally to the increased difficulty of the dataset. For example, in the CIFAR-100 experiment, our method is 1.2% better than the second best baseline (FEDAVG-PT) and 47.6% better than the worst baseline (PFEDPG).

In the TinyImageNet experiment, our method is 2.4% better than FEDAVG-PT and 50.1% better than PFEDPG. These results are therefore consistent with our findings in Table 1. Finally, Table 4 records the prompt tuning performance of the same baselines on the most challenging scenario, which combines 4 datasets from completely different vision domains. We observe a significant performance gap between PFPT and baselines that have been keeping up with its performance in earlier experiments, such as FEDAVG-PT (now 20.6% worse) and FEDPROX-PT (now 20.5% worse). More interestingly, GMM-PT, which has performed on par with our method in previous experiments, observes a significant drop in performance (now 21.4% worse). These extremely wide performance gaps are due to the fact that local prompts are more diverse (to account for highly heterogeneous tasks), which require a more refined alignment technique to facilitate effective aggregation.

### 4.2 Non-IID Data with Globally Skewed Class Distributions

In addition to the above setting which features non-IID data and locally skewed class distribution, we also evaluated the performance of our framework in a more adverse setting where the class

distribution will remain skewed even if all local datasets are aggregated. That is, local datasets in this setting are non-IID draws from a long-tailed data distribution parameterized by an imbalance factor $\text{IF} = (\max_c n_c)/(\min_c n_c)$ characterizing the ration between sample sizes ($n_c$) of the most frequent and least frequent classes [57, 58]. Such (global) long-tailed datasets are synthetically created from the original CIFAR-100 and ImageNet datasets following the data simulation in [59]. Following previous experiment setup in previous (non-ViT and no prompt-tuning) baseline methods FEDIC [57] and CReFF [58] [2], the resulting long-tailed datasets (CIFAR-100-LT and ImageNet-LT) are configured to have $\text{IF} = 10, 50, 100$ for CIFAR-100 and $\text{IF} = 1280/5$ for ImageNet. The reported results in Table 5 show that our method is also highly effective in this setting, achieving significantly higher accuracy than the current SOTA [57, 58] across all datasets and imbalance setups.

Table 5: Accuracy (%) achieved on long-tailed datasets by PFPT, CREFF [58] and FEDIC [57].

| Non-IID ($\alpha = 0.1$) | CIFAR-100-LT | | | ImageNet-LT (IF = 1280/5) |
|---|---|---|---|---|
| | IF = 100 | IF = 50 | IF = 10 | |
| FEDIC | 33.67 | 34.74 | 41.93 | 28.93 |
| CREFF | 26.19 | 28.32 | 35.49 | 26.31 |
| PFPT (ours) | **60.74** | **65.54** | **71.66** | **75.54** |

## 4.3 Prompt Convergence and Diversity

To generate insights regarding the learned prompts, we plot the 2-dimensional t-SNE embeddings of all CIFAR-100 global prompts discovered by our method across 120 communication rounds. This is repeated for the two heterogeneity and one data imbalance FL settings as well as the centralized data setting (Fig. 3). The yellow triangles in each plot mark the corresponding centroids of the prompt embeddings (one per 10 communication iterations). We also plot a best-fit spline curve (dashed and colored in red) through these centroids to visualize their update trajectory. In all settings, the distance between successive centroids consistently gets smaller as training progresses, which suggests that the learned prompts generally converge well. This is also corroborated by Fig. 4, which shows that the number of global prompts also converges over 120 communication iterations. It is also observed that in the centralized learning scenario in which no heterogeneity is present, the convergence happens much faster. The prompts quickly converge within 10 communication iterations. We also observe a gradual increase in the spread of the prompts with respect to their centroid. This suggests that the learned prompts are optimized by our method to capture the data diversity across participating clients. More interestingly, as we move from the standard heterogeneity settings (with $\alpha = 0.1$ and $\alpha = 0.5$) to the extremely heterogeneous setting with imbalance (local) data, the magnitude of this spread becomes larger, confirming the strong correlation between task heterogeneity and prompt diversity.

## 5 Conclusion

Our paper presents a new and effective approach to address the challenges of prompt-tuning pre-trained models in federated data scenarios with diverse local data distributions. Our proposed approach is a hierarchical probabilistic framework that models server aggregation as a non-parametric alignment of locally learned prompt sets into summarizing prompts. In each subsequent communication round, local clients sample and perform local updates on relevant prompts from the previous set of summarizing prompts. In this manner, our approach bypasses the effect of solution drift and outperforms existing federated learning techniques (applied on prompt-tuning) on various computer vision datasets. The reported results emphasize the cost-efficiency and efficacy of our approach in combating data heterogeneity within extremely diverse federated scenarios.

**Acknowledgements and Disclosure of Funding.** This work used GPU compute resource at SDSC through allocation CIS230391 from the Advanced Cyberinfrastructure Coordination Ecosystem: Services and Support (ACCESS) program [60], which is supported by U.S. National Science Foundation grants #2138259, #2138286, #2138307, #2137603, and #2138296. My T. Thai acknowledges the support of National Science Foundation grants SCH-2123809 and III-2416606. T.-W. Weng is supported by National Science Foundation awards CCF-2107189, IIS-2313105, IIS-2430539, the Hellman Fellowship, and Intel Rising Star Faculty Award.

---

[2]These federated fine-tuning baselines use full fine-tuning with ResNet as their backbone. Both baselines were not evaluated in our main setting in Section 4.1.

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

## A    Broader Statement of Impact

This research focuses on developing an effective prompt-tuning and summarizing algorithm for federated learning scenarios featuring a set of clients with diversely skewed local data. The mathematical approaches and insights developed in this paper will help bridge the gap between large, pre-trained models and their applications in private data settings with extremely skewed data distribution. While applications of our work to real data could result in ethical considerations, this is an indirect (and unpredictable) side-effect of our work. Our experimental work uses publicly available datasets to evaluate the performance of our algorithms; no ethical considerations are raised.

## B    Derivation of Eq. (10)

Note that $\mathbf{z}_t = \{\mathbf{z}_t^i\}_{i=1}^n$ which is a set of independent random vector given $\boldsymbol{\varphi}$. Hence,

$$\mathbb{P}\left(\mathbf{z}_t \mid \boldsymbol{\varphi}\right) \;=\; \prod_{i=1}^n \mathbb{P}\left(\mathbf{z}_t^i \mid \boldsymbol{\varphi}\right) \;=\; \prod_{i=1}^n \prod_{k=1}^{n_t} \mathbb{P}\left(z_{tk}^i \mid \boldsymbol{\varphi}\right) \tag{16}$$

where the last equality follows from the fact that $\mathbf{z}_t^i = (z_{t1}^i, z_{t2}^i, \ldots, z_{tn_t}^i)$ which is in turn a set of independent Bernoulli variables $z_{tk}^i \in \{0,1\}$ indicating whether the local prompt $\boldsymbol{\omega}_{tk}$ was sampled from a Gaussian centered at the $i$-th summarizing prompt $\phi_i$. Furthermore, following the Bernoulli process definition in Eq. (6), the $i$-th summarizing prompt $\phi_i$ is selected independently with probability $\sigma(g(\phi_i; \mathbf{w}))$ to sample the $k$-th local prompt $\boldsymbol{\omega}_{tk}$ via setting $\boldsymbol{\psi}_{tk} = \phi_i$ in Eq. (4). Hence,

$$\mathbb{P}\left(z_{tk}^i \mid \boldsymbol{\varphi}\right) \;=\; \sigma\Big(g(\phi_i; \mathbf{w})\Big)^{z_{tk}^i} \cdot \Big(1 - \sigma\Big(g(\phi_i; \mathbf{w})\Big)\Big)^{1-z_{tk}^i} \tag{17}$$

Plugging Eq. (17) into Eq. (16) and using $c_{ti} = z_{t1}^i + z_{t2}^i + \ldots + z_{tn_t}^i$ result in Eq. (10).

## C    Proof of Lemmas 3.2 and 3.3

**Lemma 3.2.** To derive the result of Lemma 3.2, note that Eq. (9) implies

$$\log \mathbb{P}\left(\boldsymbol{\omega}_t \mid \mathbf{z}_t, \boldsymbol{\varphi}\right) = \sum_{k=1}^{n_t} \log \mathbb{N}\left(\boldsymbol{\omega}_{tk} \;\middle|\; \sum_{i=1}^n z_{tk}^i \cdot \phi_i, \operatorname{diag}\left(\alpha\left(\sum_{i=1}^n z_{tk}^i \cdot \phi_i; \boldsymbol{\gamma}\right)\right)\right)$$
$$= \sum_{k=1}^{n_t} g_k\left(\sum_{i=1}^n z_{tk}^i \cdot \phi_i\right), \tag{18}$$

where we define

$$g_k\left(\mathbf{r}\right) \;\triangleq\; \log \mathbb{N}\left(\boldsymbol{\omega}_{tk} \;\middle|\; \mathbf{r}, \operatorname{diag}\left(\mathbf{r}; \boldsymbol{\gamma}\right)\right). \tag{19}$$

In addition, since $\sum_i z_{tk}^i = 1$ with $z_{tk}^i \in \{0,1\}$, Lemma 3.1 implies

$$g_k\left(\sum_{i=1}^n z_{tk}^i \cdot \phi_i\right) \;=\; \sum_{i=1}^n z_{tk}^i \cdot g_k(\phi_i), \tag{20}$$

which can be plugged into Eq. (18) to arrive at

$$\log \mathbb{P}\left(\boldsymbol{\omega}_t \mid \mathbf{z}_t, \boldsymbol{\varphi}\right) \;=\; \sum_{k=1}^{n_t} g_k\left(\sum_{i=1}^n z_{tk}^i \cdot \phi_i\right) \;=\; \sum_{k=1}^{n_t}\sum_{i=1}^n z_{tk}^i \cdot g_k(\phi_i) \tag{21}$$

$$=\; \sum_{k=1}^{n_t}\sum_{i=1}^n z_{tk}^i \log \mathbb{N}\left(\boldsymbol{\omega}_{tk} \;\middle|\; \phi_i, \operatorname{diag}\left(\phi_i; \boldsymbol{\gamma}\right)\right), \tag{22}$$

where the last equality follows from the definition of $g_k$ in Eq. 20. Finally, taking summation over $t = 1, 2, \ldots, n$ on both sides of Eq. (22) and using the definition of $\mathfrak{L}_1$, we arrive at Lemma 3.2.

**Lemma 3.3.** To derive the result of Lemma 3.3, we start with taking log on both sides of Eq. (10), which results in

$$
\log \mathbb{P}\left(\mathbf{z}_t \mid \boldsymbol{\varphi}\right) = \sum_{i=1}^{n} c_{ti} \cdot \log \sigma\Big(g\left(\boldsymbol{\phi}_i; \mathbf{w}\right)\Big) + \sum_{i=1}^{n} \Big(1 - c_{ti}\Big) \cdot \log\Big(1 - \sigma\Big(g\left(\boldsymbol{\phi}_i; \mathbf{w}\right)\Big)\Big) \quad (23)
$$

$$
= \sum_{i=1}^{n} c_{ti} \cdot \log\left(\frac{\sigma\Big(g\left(\boldsymbol{\phi}_i; \mathbf{w}\right)\Big)}{1 - \sigma\Big(g\left(\boldsymbol{\phi}_i; \mathbf{w}\right)\Big)}\right) + \sum_{i=1}^{n} \log\Big(1 - \sigma\Big(g\left(\boldsymbol{\phi}_i; \mathbf{w}\right)\Big)\Big) \quad (24)
$$

$$
= \sum_{i=1}^{n}\sum_{k=1}^{n_t} z_{tk}^i \cdot \log\left(\frac{\sigma\Big(g\left(\boldsymbol{\phi}_i; \mathbf{w}\right)\Big)}{1 - \sigma\Big(g\left(\boldsymbol{\phi}_i; \mathbf{w}\right)\Big)}\right) + \sum_{i=1}^{n} \log\Big(1 - \sigma\Big(g\left(\boldsymbol{\phi}_i; \mathbf{w}\right)\Big)\Big) \quad (25)
$$

where the last equality is due to the fact that $c_{ti} = z_{t1}^i + z_{t2}^i + \ldots + z_{tn_t}^i$. Hence, taking the summation over $t = 1, 2, \ldots, m$ on both sides of Eq. (25) and using the definition of $\mathfrak{L}_2$, we arrive at Lemma 3.3.

## D   Optimizing Eq. (15) via Iterative Weighted Bipartite Matching

For any given index $t$, Eq. (15) can be rewritten as

$$
F\Big(\mathbf{z}_t; \mathbf{z}_{-t}\Big) = \sum_{i=1}^{n}\sum_{k=1}^{n_t} z_{tk}^i \cdot \left(\log \mathbb{N}\Big(\boldsymbol{\omega}_{tk} \mid \boldsymbol{\phi}_i, \mathrm{diag}\Big(\alpha\Big(\boldsymbol{\phi}_i; \boldsymbol{\gamma}\Big)\Big)\Big) + \log\left(\frac{\sigma\Big(g\left(\boldsymbol{\phi}_i; \mathbf{w}\right)\Big)}{1 - \sigma\Big(g\left(\boldsymbol{\phi}_i; \mathbf{w}\right)\Big)}\right)\right)
$$

$$
+ \; G(\mathbf{z}_{-t}) = \sum_{i=1}^{n}\sum_{k=1}^{n_t} z_{tk}^i \cdot C_{tk}^i \;\; + \;\; G(\mathbf{z}_{-t}) \quad (26)
$$

where $G(\mathbf{z}_{-t})$ denotes the summation over the remaining terms in the definition of $L_1$ and $L_2$ that do not depend on $\mathbf{z}_t$, and

$$
C_{tk}^i \;\; \triangleq \;\; \log \mathbb{N}\Big(\boldsymbol{\omega}_{tk} \mid \boldsymbol{\phi}_i, \mathrm{diag}\Big(\alpha\Big(\boldsymbol{\phi}_i; \boldsymbol{\gamma}\Big)\Big)\Big) \;\; + \;\; \log\left(\frac{\sigma\Big(g\left(\boldsymbol{\phi}_i; \mathbf{w}\right)\Big)}{1 - \sigma\Big(g\left(\boldsymbol{\phi}_i; \mathbf{w}\right)\Big)}\right) . \quad (27)
$$

Thus, treating all but $\mathbf{z}_t$ as constants, we can solve for the optimal value of $\mathbf{z}_{-t}$ via solving the corresponding weighted bipartite matching. As we iterate over $t$, optimizing for $\mathbf{z}_t$ one at a time, this results in an iterative weighted bipartite matching approach, which is described below:

---
**Algorithm 2** Iterative Weighted Bipartite Matching

---
**input**: generative parameters $\boldsymbol{\gamma}$, $\mathbf{w}$, $\boldsymbol{\omega}$ and $\boldsymbol{\phi}$
**output**: optimized set of values for $\mathbf{z}$

1: initialize $\mathbf{z}$ randomly
2: **for** $t = 1$ **to** $m$ **do**
3:     compute the cost matrix $C_{tk}^i$ using Eq. (27) and fixing $\mathbf{z}_{-t}$ as constant
4:     $\mathbf{z}_t \leftarrow \mathrm{argmax}_{\mathbf{z}_t} F(\mathbf{z}_t; \mathbf{z}_{-t})$ // solve the weighted bipartite matching algorithm in Eq. (27)
5: **end for**
6: **return** the optimal set of alignment $\mathbf{z}$

---

## E   Additional Ablation Studies

Fig. 5 shows the learning trajectories of the summarizing prompts over 120 communication iterations of our method PFPT and GMM-PT, which is the vanilla baseline for modeling prompt alignment via clustering. Overall, the visual plots show that the learning trajectories of GMM-PT appear less

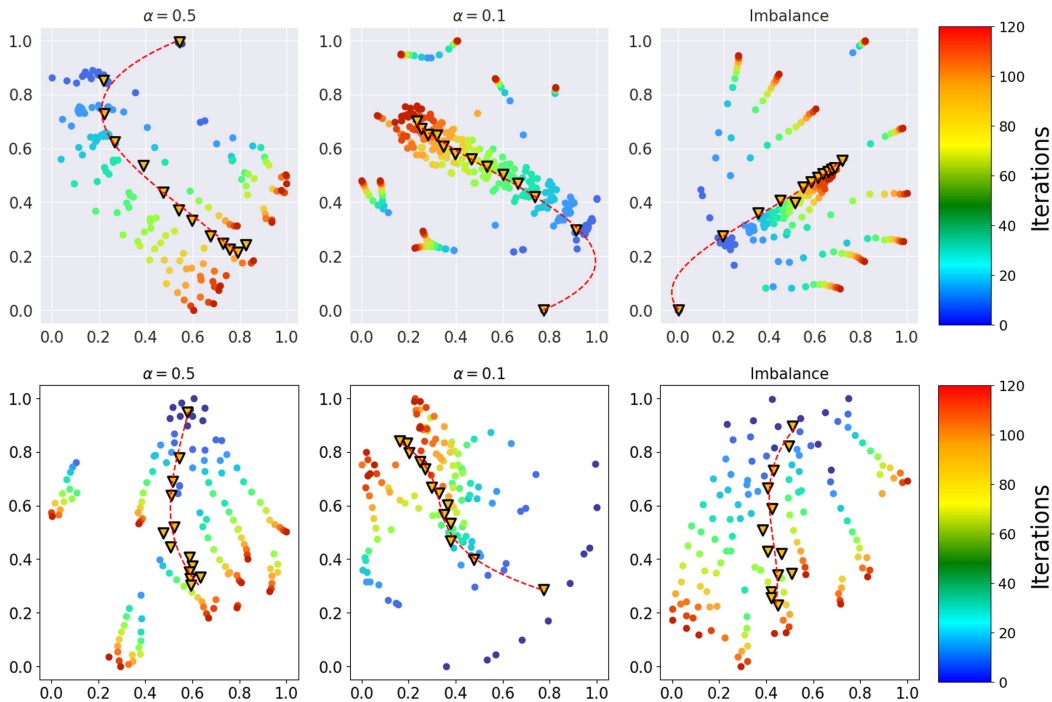

Figure 5: t-SNE plots of the (learned) summarizing prompts of (top) our method PFPT and (bottom) GMM-PT over 120 communication iterations with different heterogeneity settings. Yellow triangles denote the centroids of the t-SNE embeddings of the prompts. The dashed red lines visualize the trajectories of the centroids. The figure is best viewed with color.

diverse than those of PFPT, hinting that the better performance of PFPT can be attributed to its more diverse exploration over the prompt space, allowing it to contextualize information better into its prompt set. This appears consistent with the reported performance gap between GMM-PT and PFPT across a variety of datasets (see Table 1-Table 4 in the main text).

## F  Additional Experiment Results

In addition to the results reported in the main text, we have also conducted extra experiments comparing the performance of PFPT (ours) with a more challenging set of baselines. These are the improved variants of the original baselines we used in the main text experiment. Each baseline is now improved with a client-specific prompt selection mechanism [49], which enables each client to selectively contextualize its local data into a subset of most relevant prompts. This prevents different information contexts from being collapsed into the same prompt. This modification improves the performance of the baselines substantially on the diverse 4-dataset (Table 6) while preserving their performance on the TinyImageNet dataset (Table 7). However, it is observed that even in this more challenging setting, our PFPT still outperforms all baselines on both datasets substantially, featuring an accuracy improvement up to $4.52\%$. Similar to our observations in the main text, the improvement is most pronouncing when client data becomes more diverse, which is the case with the 4-dataset.

**Remark.** We also consider this improvement to be a part of this work's contribution (although it is not the main contribution) since the incorporation of the prompt selection mechanism in [49] has also never been investigated in the existing literature of federated fine-tuning. The comparison results in this section also helps enrich our ablation studies of the key component of PFPT.

We also run experiments to compare the performance of our PFPT algorithm against another set of federated fine-tuning baselines that incorporate adapter-tuning approach into the FedAvg and FedProx backbones. The results are reported in Table 8, which shows that the performance achieved by incorporating the adapter approach to fine-tuning in FedAvg and FedProx is only comparable

Table 6: Accuracy (%) achieved on our synthetic 4-dataset by our proposed PFPT algorithm and other baselines improved with a similar client-specific prompt selection mechanism. The symbol + indicates the improved variant of the original baseline.

|  | FEDAVG-PT+ | FEDPROX-PT+ | FEDOPT-PT+ | GMM-PT+ | PFPT |
|---|---|---|---|---|---|
| $\alpha = 0.5$ | 75.92 | 76.85 | 57.37 | 74.03 | **76.89** |
| $\alpha = 0.1$ | 66.21 | 67.77 | 61.97 | 66.47 | **70.29** |
| imbalance | 61.00 | 61.75 | 57.15 | 61.66 | **62.23** |

Table 7: Accuracy (%) achieved on the TinyImageNet dataset by our proposed PFPT algorithm and other baselines improved with a similar client-specific prompt selection mechanism. The symbol + indicates the improved variant of the original baseline.

|  | FEDAVG-PT+ | FEDPROX-PT+ | FEDOPT-PT+ | GMM-PT+ | PFPT |
|---|---|---|---|---|---|
| $\alpha = 0.5$ | 86.30 | 85.78 | 60.52 | 84.71 | **86.91** |
| $\alpha = 0.1$ | 77.79 | 75.46 | 64.60 | 76.35 | **82.31** |
| imbalance | 74.25 | 74.15 | 64.24 | 74.17 | **78.21** |

to that of PFPT on CIFAR10. Since FedAvg and FedProx were consistently the best performing baselines (after PFPT) across all datasets, we believe that the comparison of FedAvg and FedProx configured with adapter-tuning is sufficient to demonstrate the advantage of PFPT over potential adapter-based FL approaches.

Table 8: Accuracy (%) achieved on the all dataset by FEDAVG with Adapter-Tuning, FEDPROX with Adapter-Tuning, and our proposed PFPT algorithm.

| Method | Setting | CIFAR10 | CIFAR100 | TinyImageNet | synthetic 4-datasets |
|---|---|---|---|---|---|
| FEDAVG-Adapter | $\alpha = 0.5$ | 93.86±0.17 | 75.95±0.40 | 78.88±0.23 | 55.55±0.79 |
|  | $\alpha = 0.1$ | 92.66±0.26 | 65.04±0.68 | 57.62±0.80 | 30.58±4.67 |
|  | imbalance | **92.33±0.26** | 49.8±0.79 | 40.90±1.34 | 10.86±8.94 |
| FEDPROX-Adapter | $\alpha = 0.5$ | 93.69±0.20 | 75.75±0.16 | 79.01±0.56 | 58.39±1.15 |
|  | $\alpha = 0.1$ | 93.04±0.33 | 64.59±0.82 | 58.62±0.56 | 32.92±1.34 |
|  | imbalance | 92.13±0.13 | 50.75±1.71 | 37.65±2.19 | 13.19±7.92 |
| PFPT | $\alpha = 0.5$ | **94.39±0.51** | **80.24±0.24** | **86.91±0.14** | **76.89±0.17** |
|  | $\alpha = 0.1$ | **93.39±0.22** | **75.08±0.51** | **82.31±0.26** | **70.29±0.32** |
|  | imbalance | 91.45±0.08 | **72.05±0.93** | **78.21±1.25** | **62.23±1.02** |

# G  Hyperparameter Settings

All experimented baselines use a batch size of 16 and are fine-tuned with 10 learnable prompts. All experiments are performed on a V100 GPU with 32GB GPU RAM. We employ the same hyperparameter setting in all baselines (except for SCAFFOLD-PT). The hyperparameter settings are all presented in Table 9. SCAFFOLD-PT requires a larger learning rate since SCAFFOLD's variate control mechanism is designed to work with SGD. Hence, unlike other baselines, SCAFFOLD-PT is configured with SGD instead of Adam. Empirically, we find that a larger learning rate is required for SCAFFOLD-PT with SGD to reach its best performance in our setting.

# H  Limitations

Despite the clear advantage of PFPT in terms of communication bandwidth, as opposed to full-model FT and other forms of FL that typically have to communicate the entire architecture weights per round of communication, we foresee that PFPT will face some challenges when the pre-trained and downstream task domains differ (which goes even beyond the distribution shift setting that we investigate in this paper). Intuitively, a significant domain shift will likely require using more trainable prompts. Can we quantify this shift and bound the minimum number of prompts required to enable accurate adaptation? Although this question is beyond the scope of this paper, we will investigate

Table 9: Hyperparameter setting for all baselines and our PFPT

| Method | Setting | Batch size | Communication round | Eps. in local training | Optimizer & learning rate | Total clients | Sampled clients |
|---|---|---|---|---|---|---|---|
| FEDAVG-PT FEDPROX-PT FEDAVG-Adapter FEDPROX-Adapter FEDOPT-PT PFEDPG-PT GMM-PT PFPT | $\alpha = 0.5$ | 16 | 120 | 5 | Adam: beta=(0.9, 0.98), eps=1e-6 lr: 5e-4 | CIFAR10: 100 CIFAR100: 100 TinyImageNet: 100 Synthetic 4-dataset: 80 | 10 |
| | $\alpha = 0.1$ | 16 | 120 | 5 | Adam: beta=(0.9, 0.98), eps=1e-6 lr: 1e-4 | CIFAR10: 100 CIFAR100: 100 TinyImageNet: 100 Synthetic 4-dataset: 80 | 10 |
| | imbalance | 16 | 120 | 5 | Adam: beta=(0.9, 0.98), eps=1e-6 lr: 1e-4 | CIFAR10: 100 CIFAR100: 100 TinyImageNet: 100 Synthetic 4-dataset: 80 | 10 |
| SCAFFOLD-PT | $\alpha = 0.5$ | 16 | 200 | 5 | SGD lr: 5e-4 | CIFAR10: 100 CIFAR100: 100 TinyImageNet: 100 Synthetic 4-dataset: 80 | 10 |
| | $\alpha = 0.1$ | 16 | 200 | 5 | SGD lr: 1e-4 | CIFAR10: 100 CIFAR100: 100 TinyImageNet: 100 Synthetic 4-dataset: 80 | 10 |
| | imbalance | 16 | 200 | 5 | SGD lr: 1e-4 | CIFAR10: 100 CIFAR100: 100 TinyImageNet: 100 Synthetic 4-dataset: 80 | 10 |

and address it in future works, especially in the context of heterogeneous FL where each client operates in a different domain. We have also raised the issue of prompt tuning having to compute all intermediate gradients since the prompt tokes are appended at the top level of the pre-trained model. Going forward, we can explore new prompt placement strategies to reduce this memory consumption.

