# OpenReview forum: "Probabilistic Federated Prompt-Tuning with Non-IID and Imbalanced Data"
_NeurIPS.cc/2024/Conference — NeurIPS 2024 poster_

### Official Review · Reviewer_HV8G · 2024-07-07

**Soundness:** 3
**Presentation:** 3
**Contribution:** 3
**Rating:** 7
**Confidence:** 4

**Summary:**

Mainly in the setting of imbalanced data, this paper proposes a probabilistic federated prompt-tuning pre-trained model from two aspects: local prompt generation and global prompt aggregation. In local prompt generation, each local set is assumed to be a random set of prompts distributed by a hierarchical generative model. In global prompt aggregation, non-parametric algorithm aligns local prompts of similar clients. On public computer vision datasets, experiment results demonstrate the effectiveness of the proposed method.

**Strengths:**

1. This method is interpretable, and the local prompt is obtained based on Bernoulli distribution and Gaussian distribution sampling, and the EM algorithm is used to optimize the iterative process.
2. The quantitative results of imbalanced data and the prompts embedding results of t-SNE can fully show the effectiveness of the proposed method.

**Weaknesses:**

1.The overall framework of the method is not clear enough. Although there is an overall framework diagram in the appendix, it omits a lot of details. It is confusing that the two DNNS are on the server side, and are the two DNNS on different clients consistent?
2.The quantitative comparison results of this method show GMM-PT, but the advantages of the proposed method and the theoretical distinction can be further explained.

**Questions:**

1. The specific implementation details of the model, such as the DNN employed, the specific configuration of the pre-trained model seem to be unclear?
2. The lack of discussion of federated prompt-tuning pre-trained models in related work seems to be very relevant to this paper.

**Limitations:**

The lack of a discussion of limitations and deficiencies is important.

---

> ### Author Rebuttal · Authors · 2024-08-07
>
> We would like to thank the reviewer for recognizing our contribution.
> Your questions are addressed below:
>
> **Q1. Workflow Diagram.** We apologize for the confusion. Our workflow diagram illustrates two different phases of our algorithm as annotated in the caption of Fig. 4. These two phases are iterative and thematically separated by the purple arrow:
>
> On the left, the diagram highlights the local phase where each client (1) pulls a subset of prompts from the global set of the previous iteration and then (2) fine-tunes them.
>
> On the right, the diagram highlights the global phase where local clients send their prompt sets to the server which then aggregates them.
>
> **Q2. Implementation Detail.** Our work is based on a pre-trained Vision Transformer (ViT_B_32 pre-trained by ImageNet from PyTorch [a]. Each local client will fine-tune the pre-trained model using a local set of learnable prompts. At the end of each iteration, the local prompt sets will be shared with the server which then aggregates them into a global prompt set. Different baselines use different aggregation algorithms (see Appendix G).
>
> [a] https://pytorch.org/vision/main/models/generated/torchvision.models.vit_b_32.html.
>
> **Q3. Discussion of federated prompt-tuning pre-trained models.** We have provided this discussion in the introduction at lines 42-50. As the existing literature on federated prompt-tuning is relatively few [21-23] with most existing work utilizing FedAvg to aggregate local prompt sets, ignoring the prompt alignment issue, we decided to merge this discussion with our positioning in the introduction. If this work is accepted, we can use the 10th page to expand this discussion.
>
> We hope our response has addressed your questions. Please let us know if you have any follow-up questions for us.

---

> > ### Comment · Reviewer_HV8G · 2024-08-08
> >
> > Thanks for the response. The authors have addressed my concerns, and I will raise my score to accept.

---

> > > ### Author Response · Authors · 2024-08-08
> > > **Thank you for raising the score to accept**
> > >
> > > Dear Reviewer HV8G,
> > >
> > > Thank you very much for agreeing to raise the score from weak accept to accept. We find this very encouraging!
> > >
> > > We would appreciate it if you could also update the score in the original review.
> > >
> > > Best regards,
> > >
> > > Authors

---

### Official Review · Reviewer_h6E3 · 2024-07-09

**Soundness:** 2
**Presentation:** 3
**Contribution:** 2
**Rating:** 5
**Confidence:** 5

**Summary:**

This paper studies the problem of prompt tuning in FL to address the data imbalance problem. The topic is interesting and broad enough for the community. The motivation and problem setting are good and promising. Experiments are sufficient to support the effectiveness of the proposed method.

**Strengths:**

1.	The problem of heterogeneous and imbalanced FL is interesting.
2.	The overall writing and structure of this paper is good.
3.	The experiments are sufficient to support the effectiveness of the proposed method.

**Weaknesses:**

1.	The local imbalance setting is very extreme, as described in section 4.1. There’s no real-world application to support this setting.
2.	This paper lacks discussion of existing FL methods that address the class imbalance problem like:
[1] Xinyi Shang, Yang Lu, Gang Huang, and Hanzi Wang, “Federated Learning on Heterogeneous and Long-Tailed Data via Classifier Re-Training with Federated Features,” in IJCAI, 2022.
[2] Xinyi Shang, Yang Lu, Yiu-ming Cheung, and Hanzi Wang, “FEDIC: Federated Learning on Non-IID and Long-Tailed Data via Calibrated Distillation,” in ICME, 2022.
[3] Wenke Huang, Yuxia Liu, Mang Ye, Jun Chen, Bo Du, “Federated Learning with Long-Tailed Data via Representation Unification and Classifier Rectification” in TIFS, 2024.
Although the addressing problem is different in that the class imbalance is global, there are still connections in terms of problem settings.
3.	The proposed prompt aggregation method seems not to directly address the problem of data imbalance. It is a more general solution to address the problem of heterogeneous FL. From the experiments, it can also be observed that PFPT achieves better performance on both non-iid and imbalance scenarios.
4.	In the experiment, there should be more imbalance settings to support the robustness of PFPT. Currently, there is only one setting.

**Questions:**

1.	What are the real-world applications to support the scenario of extreme local imbalance in FL?
2.	How does PFPT explicitly solve the problem of data imbalance?
3.	Global imbalance seems more common in real-world applications. Can PFPT be effective in addressing the global imbalance problem?

---

> ### Author Rebuttal · Authors · 2024-08-07
>
> We would like to thank the reviewer for the detailed questions which we have addressed below.
>
> **Q1. Real-world application to support this setting.** We would like to emphasize that our work has first showcased its improved performance on standard non iid settings simulated with Dirichlet prior -- see the first 2 rows of Tables 1-4 -- on local class distribution which is the common practice in prior heterogeneous FL work (including the references [1-3] suggested by the reviewer). This already implies a certain degree of imbalance in local data distribution. The extreme imbalance setting is considered as a limit test in the worst case. Previous works have also studied such setting [a-f]. Extreme class imbalance often happens in practical scenarios, e.g.,
> (a) rare disease detection with extremely low prevalence rate (less than 1/10000) [e]; and
> (b) mobile keyboard prediction where on-device datasets are often imbalanced, with a few classes (e.g., commonly typed words or phrases) dominating [f].
>
> [a] FedIIC: Towards Robust FL for Class-Imbalanced Medical Image Classification (https://arxiv.org/abs/2206.13803)
>
> [b] FL with Non-IID Data (https://arxiv.org/abs/1806.00582)
>
> [c] Addressing Class Imbalance in FL (https://arxiv.org/abs/2008.06217)
>
> [d] FL with Matched Averaging (https://www.arxiv.org/abs/2002.06440)
>
> [e] Complementary Pattern Augmentation for Rare Disease Detection (https://arxiv.org/abs/1911.13232)
>
> [f] FL for Mobile Keyboard Prediction (https://arxiv.org/abs/1811.03604)
>
>
> **Q2.**
>
> **A. Discussion of existing FL methods addressing class imbalance**
>
> We thank the reviewer for suggesting these additional references [1-3]. We note that these works focus on full fine-tuning approaches in settings where the centralized dataset is imbalanced. Instead, we focus on light-weight prompt-tuning scenarios in which the local datasets are imbalanced but their centralization would be reasonably balanced. As such, our setting and the settings adopted in these additional references are complementary. We will cite them and include the above discussion in our revision.
>
> **B. Explain how PFPT solves the problem of (local) data imbalance.**
>
> Local data imbalance will cause diversity across local prompt sets (i.e., clients). We refer the reviewer to Fig. 2, which shows that the local prompts indeed diverge to capture different data patterns. This means the same prompt position across different clients might encode different context information of the fine-tuning task. A simple aggregation that simply combines prompts at the same position across different clients from different contexts might therefore collapse them into less informative prompts.
> To avoid this, we must learn a prompt alignment so that we only aggregate prompts which encode the same aspect of context information. PFPT models this prompt alignment as a parameter of our generative model of the local prompts. Maximizing the observation likelihood of the local prompts will allow us to learn the most plausible alignment, hence solving the local data imbalance issue.
>
> **Re: the concern that the proposed prompt aggregation method is a more general solution to address the problem of heterogeneous FL and achieves better performance on both non-iid and imbalance scenarios:**
>
> We are not sure if this is considered a weakness. However, we agree that the title of our paper might be a bit too specific on imbalance data settings. Our main point is that existing prompt aggregation performs worse when the local data become more diverse and imbalanced. It can be observed (from all table) that the performance improvement over baselines increases when the data becomes more imbalanced. In the synthetic 4-dataset benchmark, with more diverse data collected from 4 different datasets, the performance gain over the baselines is much more pronounced.
>
> **Q3. More imbalanced settings & how effective PFPT is in global imbalance**
>
> Our work focuses on the common setting in existing heterogeneous FL work where the centralized dataset would be reasonably balanced but each local dataset can be imbalanced due to the client heterogeneity. This is the main cause of the solution drift issue.
>
> However, to our understanding, we believe the reviewer wants us to explore the settings under which the centralized dataset is imbalanced (i.e., global imbalance). This presents an orthogonal challenge to the solution drift challenge considered in existing heterogeneous FL literature (including ours) which is beyond the current scope and deserves a separate treatment.
>
> Nonetheless, we have run additional experiments comparing with the methods in the references [1-3] that the reviewer suggested. The experiments also use the same long-tailed datasets & FL scenarios detailed in those references. The results show that our method achieves higher accuracy than prior works substantially across all global imbalance settings with different imbalance factors (IF):
>
> | CIFAR100-LT ($\alpha=0.1$) | IF = 100 | IF = 50 | IF = 10 |
> |-------------------------|----------|---------|---------|
> | FEDIC [2]  | 33.67    | 34.74   | 41.93   |
> | PFPT (ours) | 60.74    | 65.54   | 71.66   |
>
> | CIFAR100-LT ($\alpha=0.5$) | IF = 100       | IF = 50        | IF = 10        |
> |-------------------------|----------------|----------------|----------------|
> | CReFF [1]  | 34.67 | 37.64 | 47.08 |
> | RUCR [3]                 | 36.83          | 40.80          | 50.90          |
> | PFPT (ours)             | 60.69          | 65.41          | 73.68          |
>
> | ImageNet-LT ($\alpha=0.1$) | |
> |-----------------------------------|:---------------:|
> | CReFF [1]                          |      26.31      |
> | FEDIC  [2]                         |      28.93      |
> | PFPT (ours)                       |      75.54      |
>
> We hope the reviewer will consider increasing the rating if our response has addressed the questions sufficiently. We will be happy to answer any follow-up questions that the reviewer might have for us.

---

> > ### Comment · Reviewer_h6E3 · 2024-08-08
> >
> > Thanks for the authors' response. Most of my concerns are solved. I'm willing to increase my score to 5. However, I still suggest the authors reconsider the title because "data imbalance settings" are too broad to be covered by the proposed method.

---

> > > ### Author Response · Authors · 2024-08-08
> > > **Thank you for increasing the rating**
> > >
> > > Dear Reviewer h6E3,
> > >
> > > Thank you for the fast response. We appreciate the rating increase!
> > > We will update the title in our revision.
> > >
> > > Best regards,
> > >
> > > Authors

---

> ### Author Response · Authors · 2024-08-07
> **Re: Additional detail regarding the extra experiments we run in response to the reviewer's Q3**
>
> Please note that not all methods in those references are tested on all those FL scenarios. Hence, for each scenario (reported in a separate table -- see Q3 in the main rebuttal), we only compare with the (best) reported performance of the methods which were tested on that scenario.
>
> --
>
> All local data distributions and FL settings are configured to be the same for fair comparison:
> Batch size=32 for $\alpha=0.5$, 128 for $\alpha=0.1$ and 32 for ImageNet-LT.
>
> No. total clients=20.
>
> No. online clients in each communication round=8.
>
> No. epochs in local training=10.
>
> --
>
> We used the following released implementation of those references:
>
> FEDIC: https://github.com/shangxinyi/FEDIC/blob/main/options.py
>
> CReFF-FL: https://github.com/shangxinyi/CReFF-FL
>
> RUCR: https://github.com/liuyuxia211/RUCR/blob/main/options.py
>
> --
>
> [1] Xinyi Shang, Yang Lu, Gang Huang, and Hanzi Wang, “Federated Learning on Heterogeneous and Long-Tailed Data via Classifier Re-Training with Federated Features,” in IJCAI, 2022.
>
> [2] Xinyi Shang, Yang Lu, Yiu-ming Cheung, and Hanzi Wang, “FEDIC: Federated Learning on Non-IID and Long-Tailed Data via Calibrated Distillation,” in ICME, 2022.
>
> [3] Wenke Huang, Yuxia Liu, Mang Ye, Jun Chen, Bo Du, “Federated Learning with Long-Tailed Data via Representation Unification and Classifier Rectification” in TIFS, 2024

---

### Official Review · Reviewer_tCnK · 2024-07-13

**Soundness:** 2
**Presentation:** 3
**Contribution:** 3
**Rating:** 6
**Confidence:** 3

**Summary:**

This paper introduces a novel approach to prompt-tuning within the federated learning (FL) framework, focusing on enhancing the adaptability of pre-trained models across diverse clients using a probabilistic method. They proposed a hierarchical approach to model the generation and aggregation of local prompts. They develop a way to associate local (client side) and summarizing prompts (server side), utilizing a weighted bipartite matching task that linearly interacts with the model's loss function to optimize prompt association. Their experimental results show that the proposed method is effective in data imbalance settings.

**Strengths:**

- The authors introduce an innovative probabilistic approach to prompt tuning within the context of Federated Learning.
  - The logic of this paper is clear and easy to follow.
  - The insights provided in the paper have the potential to inspire further research within the community, suggesting directions for integrating probabilistic approaches into federated tuning.

**Weaknesses:**

- The paper primarily focuses on prompt tuning; a comparative analysis with other efficient tuning methods would strengthen its persuasiveness.
  - In Sec. 2, the related work mainly focuses on the solution drift issue, yet it lacks a comprehensive discussion on various efficient tuning methods under scenarios of data scarcity. The authors need to provide additional motivation for focusing on prompt tuning and expand on related work concerning prompt tuning.
  - The paper addresses data imbalance in Federated Learning but provides limited evidence supporting the proposed method. More substantial theoretical or experimental evidence is needed to demonstrate its effectiveness in imbalance settings.

**Questions:**

1. In Sec. 3.1's remarks, an alternative tuning strategy, adapter networks, is mentioned. Could the authors clarify the advantages of their proposed method over these adapter network approaches?
  2. In Sec. 3.2.3, I think the initialization of summarizing prompts will affect the performance. Could the authors specify how these prompts were initialized in the experiments?
  3. In lines 178-181, it is hypothesized that each prompt captures a specific pattern or concept. Could the authors provide evidence or further explanation to support this assumption? This clarification would strengthen the theoretical foundation of the study.
  4. In lines 223-225, could the authors provide more evidence for why within a single client, at most only one local prompt would be associated with the i-th summarizing prompt?
  5. In Appendix G, "10 learnable prompts" are mentioned. Could the authors explore whether increasing this number could enhance the performance of the proposed method? More prompts could potentially allow for the learning of finer-grained concepts.
  6. In line 201, could the authors provide more details on the method each client uses to select a subset of summarizing prompts?

---

> ### Author Rebuttal · Authors · 2024-08-07
>
> We thank the reviewer for recognizing our contribution and for the constructive feedback, which are addressed below.
>
> **Q1. Discussing other fine-tuning approaches.**
>
> We agree with the reviewer that a more comprehensive discussion on PFPT will make our contribution position clearer. However, we want to note that our main contribution is about addressing a technical issue (i.e., prompt alignment) of an existing federated prompt-tuning approach, which is essential in settings with heterogeneous and/or imbalance local data:
>
> --
>
> Existing federated prompt-tuning approaches [21-23] have ignored the important issue of prompt alignment across different clients. Aggregating misaligned prompts could result in suboptimal performance under imbalanced and/or heterogeneous local data (see our experiments). Our work aims to fix this problem. Therefore, our focus is necessarily specific to prompt-tuning.
>
> --
>
> Generalizing our proposed solution to other fine-tuning techniques and finding which one is most effective is indeed interesting but orthogonal to our contribution. We will investigate this in a separate follow-up work.
>
> Nonetheless, we have provided a comparative analysis with adapter tuning in Appendix F (Table 7), which shows that prompt-tuning outperforms adapter-tuning (FEDAVG-Adapter, FEDPROX-Adapter) in most FL scenarios on 4 benchmark datasets. The results in Table 7 are quoted below and also supplemented with additional comparison with two other variants of federated adapter-tuning (FEDOPT-Adapter, SCAFFOLD-Adapter). We note that these new baselines were created during the rebuttal week for a more thorough comparison but they have not been investigated in prior literature.
>
> | $\alpha=0.5$ | CIFAR10 | CIFAR100 | TinyImageNet | synthetic 4-dataset |
> |------------------|----------------|----------------|----------------|---------------------|
> | FEDAVG-Adpater | 93.86±0.17 | 75.95±0.40 | 78.88±0.23 | 55.55±0.79 |
> | FEDPROX-Adapter | 93.69±0.20 | 75.75±0.16 | 79.01±0.56 | 58.39±1.15 |
> | FEDOPT-Adapter | 89.34±0.77 | 35.96±1.32 | 23.51±0.65 | 31.85±0.85 |
> | SCAFFOLD-Adapter | 78.10±0.39 | 18.64±1.54 | 22.41±0.65 | 29.72±0.71 |
> | PFPT (Ours) | **94.39±0.51** | **80.24±0.24** | **86.91±0.14** | **76.89±0.17** |
>
> | $\alpha=0.1$ | CIFAR10 | CIFAR100 | TinyImageNet | synthetic 4-dataset |
> |------------------|----------------|----------------|----------------|---------------------|
> | FEDAVG-Adpater | 92.66±0.26 | 65.04±0.68 | 57.62±0.80 | 30.58±4.67 |
> | FEDPROX-Adapter | 93.04±0.33 | 64.59±0.82 | 58.62±0.56 | 32.92±1.34 |
> | FEDOPT-Adapter | 85.32±1.40 | 22.07±3.03 | 14.76±1.06 | 20.01±1.82 |
> | SCAFFOLD-Adapter | 79.06±0.81 | 12.27±1.76 | 17.87±1.19 | 18.68±2.44 |
> | PFPT (Ours) | **93.39±0.22** | **75.08±0.51** | **82.31±0.26** | **70.29±0.32** |
>
> | Imbalance | CIFAR10 | CIFAR100 | TinyImageNet | synthetic 4-dataset |
> |------------------|----------------|----------------|----------------|---------------------|
> | FEDAVG-Adpater | **92.33±0.26** | 49.8±0.79 | 40.90±1.34 | 10.86±8.94 |
> | FEDPROX-Adapter | 92.13±0.13 | 50.75±1.71 | 37.65±2.19 | 13.19±7.92 |
> | FEDOPT-Adapter | 74.22±1.11 | 11.75±3.64 | 10.16±1.00 | 21.13±5.20 |
> | SCAFFOLD-Adapter | 80.56±1.08 | 20.56±2.12 | 22.49±1.79 | 4.33±1.91 |
> | PFPT (Ours) | 91.45±0.08 | **72.05±0.93** | **78.21±1.25** | **62.23±1.02** |
>
> In our revision, we will also include a broader discussion of existing fine-tuning techniques in Section 2, which is summarized in an additional comment below (due to limited rebuttal space).
>
> **Q2. Initialization of summarizing prompts.**
>
> The initialization of prompts would not affect the performance as it follows the standard initialization in [a] which is commonly used in deep learning. The reported std in our experiments is also relatively small, indicating that prompt initialization has little effect on performance variation.
>
> [a] Understanding the difficulty of training deep feedforward neural networks. AISTATS (2010).
>
> **Q3. Each prompt captures a specific pattern of concept.**
>
> We refer the reviewer to Fig. 2 in our paper, which shows the t-SNE plots of the (learned) summarizing prompts on CIFAR-100 over 120 communication iterations. Yellow triangles denote the centroids of the t-SNE embeddings of the prompts. The dashed red line visualizes their trajectories. The plots show that each prompt follows a different trajectory, suggesting that each prompt does capture a specific pattern or concept.
>
> **Q4. For a single client, at most only one local prompt would be associated with the i-th summarizing prompt?** This is due to the generative design in Section 3.2.1 (line 183-186). Per client, each summarizing prompt will flip a Bernoulli coin with learnable probability to decide whether to sample one local prompt from its vicinity. If the decision is positive, the sampled local prompt is said to be associated with the summarizing prompt. Hence, per client, for each summarizing prompt, there is at most one associated local prompt. This setup is similar to that of the setting in The Indian Buffet Process: An Introduction and Review. JMLR (2011)
>
> **Q5. More prompts improve performance.** We have run additional experiments showing the increasing performance of our model with an increasing number of prompts -- see the figure in the attached PDF in our summary response.
>
> **Q6. Method that each client uses to select a subset of summarizing prompts?**
> We adopt the prompt selection mechanism in “Learning to Prompt for Continual Learning” (CVPR, 2022)
>
> We hope the reviewer will consider increasing the rating if our response has addressed the questions sufficiently. We will be happy to answer any follow-up questions that the reviewer might have for us.

---

> > ### Author Response · Authors · 2024-08-09
> > **Follow-up**
> >
> > Dear Reviewer tCnK,
> >
> > Thank you again for the detailed feedback. We hope our responses have addressed your questions sufficiently.
> > We are happy to discuss further if you have follow-up questions for us.
> >
> > Best regards,
> >
> > Authors

---

> > > ### Author Response · Authors · 2024-08-12
> > > **Re: Follow-up**
> > >
> > > Dear Reviewer tCnK,
> > >
> > > May we know if our response has addressed your questions sufficiently?
> > >
> > > We really appreciate your detailed feedback, which (as addressed above) will be incorporated into our revision.
> > >
> > > Best regards,
> > >
> > > Authors

---

> > > > ### Comment · Reviewer_tCnK · 2024-08-13
> > > > **Thanks for the nice rebuttal**
> > > >
> > > > Thanks for the author’s responses and additional experiments. Most of the concerns are addressed. And I will increase the score to 6.

---

> > > > > ### Author Response · Authors · 2024-08-13
> > > > > **Re: Thank you for increasing the rating**
> > > > >
> > > > > Dear Reviewer tCnK,
> > > > >
> > > > > Thank you for increasing the rating!
> > > > >
> > > > > We are glad our response has addressed your concerns.
> > > > >
> > > > > Best regards,
> > > > >
> > > > > Authors

---

> ### Author Response · Authors · 2024-08-07
> **Extended discussion on fine-tuning approaches (supplement content to our main rebuttal)**
>
> Due to the limited rebuttal space, we defer this editorial discussion content to this separate comment:
>
> --
>
> Existing fine-tuning approaches either use prompts to adapt the input [a] or adapter networks [b] to adapt the pre-trained weights. Both of which will help modify the pre-trained model to fit the context of a downstream task.
>
> Prompt-tuning methods focus on engineering cues, such as extra tokens which are appended as prefixes to the sequence of input embeddings to a multi-head self-attention unit. Such tokens or prompts provide beneficial context to performing the computational task, similar to how hints can be provided to assist puzzle solving.
>
> Adapter tuning is a parameter-efficient method for fine-tuning large foundation models. Instead of updating all the parameters of a model, adapter tuning involves adding small, trainable modules (adapters) to the model. During training, only these adapters are updated, while the rest of the model’s parameters remain fixed.
>
> The fundamental difference between these two approaches is that prompt-tuning only modifies the contextual information in the query, whereas adapter tuning instead alters how the pre-trained model behaves. In centralized learning, prompt-tuning is more memory efficient as the sizes of the prompts at the input embedding level only depend on the input sizes while the sizes of the update still depend on the model sizes. Thus, in the context of FL with resource-limited devices (e.g., wearable devices for health monitoring), prompt-tuning is more affordable in terms of memory usage. On the other hand, adapter networks can perform intricate updates to the model, and thus are suitable for more complex tasks.
>
> Both fine-tuning approaches are less investigated in the federated settings with heterogeneous and imbalance data. A few recent works [21-23] (as cited in our main text) have investigated a potential integration between FedAvg and prompt-tuning but have not addressed the prompt alignment issue (as elaborated in lines 51-59 in our main text) and have also not considered the heterogeneous and imbalance data setting.
>
> [a] "The power of scale for parameter-efficient prompt tuning." EMNLP (2021).
>
> [b] “Learning multiple visual domains with residual adapters”. NeurIPS (2017).

---

### Official Review · Reviewer_DxxX · 2024-07-13

**Soundness:** 3
**Presentation:** 3
**Contribution:** 3
**Rating:** 5
**Confidence:** 3

**Summary:**

This paper address the challenges of prompt-tuning pre-trained models in federated learning scenarios with diverse local data distributions. Specifically, it formulates the prompt summarizing procedure as a probabilistic set modeling task, treating each local set as an independent sample of a random point process and aligning similar prompts across different sets as part of the modeling parameterization. The research compares the proposed method's performance against various federated prompt-tuning baselines, demonstrating its effectiveness in combating data imbalance in extremely heterogeneous scenarios through a series of experiments and evaluations.

**Strengths:**

1. The paper introduces a novel probabilistic set modeling approach for prompt summarization, which enhances the understanding and processing of local sets as independent samples of a random point process.
2. The method is good in addressing data imbalance in extreme heterogeneous scenarios, which is a significant challenge in federated learning environments.
3. The research includes comprehensive experiments and comparisons against existing federated prompt-tuning baselines, providing robust evidence of the method’s effectiveness.
4. The use of classical weighted bipartite matching within the generative model’s framework adds a layer of theoretical rigor to the research, grounding the practical contributions in solid mathematical foundations.

**Weaknesses:**

1. “Papers to be submitted to NeurIPS 2024 must be prepared according to the instructions presented here. Papers may only be up to nine pages long, including figures. Additional pages containing only acknowledgments and references are allowed.” There is a minor formatting violation.

**Questions:**

-

**Limitations:**

-

---

> ### Author Rebuttal · Authors · 2024-08-07
>
> We would like to thank the reviewer for recognizing the strength of our paper and for helping us catch a format issue.
>
> We thought the broader impact section is counted as part of the checklist content which is not counted towards the page limit. It is possible that we had misunderstood the policy regarding the broader impact statement. We will move this section to the appendix in the revision to make sure it is not considered part of the main text.
>
> We hope the reviewer will consider increasing the rating of our paper as this minor format issue is the only concern and it does not impact the technical contribution of our work.
>
> Thank you for your consideration.

---

> > ### Comment · Reviewer_DxxX · 2024-08-08
> > **Rating**
> >
> > Given the authors’ response and other reviews' comments, I will maintain my rating.

---

### Author Rebuttal · Authors · 2024-08-07

We thank all reviewers for their constructive comments. We summarize below our responses to the reviewers’ questions and concerns, as well as additional results to support our method.

**Reviewer HV8G** requested several clarifications and minor adjustments of our manuscript, which we have thoroughly addressed.

**Reviewers tCnK and h6E3** requested additional discussion regarding other fine-tuning techniques and FL techniques that deal with class imbalance, which we have provided in the respective responses. We have also conducted extra experiments to compare our method with these techniques. We refer the reviewers to our response to **Q1 of reviewer tCnK** for an empirical comparison with adapter tuning techniques, and to **Q3 of reviewer h6E3** for another empirical comparison with FL techniques handling data imbalance (in the global settings suggested by the references provided by **reviewer h6E3**). These additional results show that our method performs robustly against other baselines across different settings.

**Reviewer tCnK** requested additional experiment showing that the model performance will increase with a larger no. of prompts (Q5). We have provided that experiments in the attached PDF.

**Reviewer h6E3** also requested for evidence that our experiment setting is realistic. We would like to highlight that all of our Dirichlet partitioning scenarios are standard in many previous FL work. Our extreme imbalance partitioning scenario has also been investigated in several well-cited works, which we have provided references in our response to **Q1 of Reviewer h6E3**.

**Reviewer DxxX** raised a format issue with the broader impact statement which will be fixed in our revised draft.

We thank all reviewers again for the constructive reviews and we hope our response has sufficiently addressed all questions. We will be happy to answer any follow-up questions that the reviewers might have during the Reviewer-Author discussion week.

---

### Author Response · Authors · 2024-08-14
**Post-rebuttal summary**

Dear AC and Reviewers,

We would like to express our gratitude to the Area Chair for coordinating the review process of our paper.

We are very glad that all reviewers found our rebuttal satisfactory and have given acceptance scores.

We would also like to thank Reviewer DxxX for maintaining the original acceptance rating and Reviewers tCnK, h6E3, and HV8G for increasing their ratings of our work.

We really appreciate your timely feedback and will revise the paper accordingly to incorporate our post-rebuttal discussion.

Best regards,

Authors

---

### Decision · Program_Chairs · 2024-09-25

**Decision:**

Accept (poster)

**Comment:**

The proposed method is novel and practical. All reviewers are positive to accept the paper. It could be further enhanced by justifying the relation between the proposed method and the proposed scenario.